# Determination of the Kinematic Excitation Originating from the Irregular Envelope of an Omnidirectional Wheel

**DOI:** 10.3390/s21206931

**Published:** 2021-10-19

**Authors:** Sławomir Duda, Olaf Dudek, Grzegorz Gembalczyk, Tomasz Machoczek

**Affiliations:** 1Department of Theoretical and Applied Mechanics, Faculty of Mechanical Engineering, Silesian University of Technology, 44-100 Gliwice, Poland; Grzegorz.Gembalczyk@polsl.pl (G.G.); Tomasz.Machoczek@polsl.pl (T.M.); 2ETISOFT Smart Solutions Sp. z o.o., 44-100 Gliwice, Poland; odudek@ess.etisoft.pl

**Keywords:** omnidirectional wheel, vibration measurement system, mobile robots

## Abstract

This paper describes a test stand for determining the kinematic excitation originating from the contact between a vehicle’s wheel and the ground, thus acting on the single suspension upright of the vehicle. This excitation is unique to the movement of omnidirectional wheels and originates from the irregular envelope of the wheel. The presented attitude enables the vertical displacement of the wheel’s axis rolling on a horizontal surface to be determined. This work includes experimental results considering different wheel orientations against the direction of movement.

## 1. Introduction

The movement of a wheeled vehicle on an irregular surface typically results in the excitation of the vehicle’s body, causing vibration. The vibration arises from the impact of surface irregularities on the wheel suspension system and the vehicle body, thus leading to discomfort while driving and poor vehicle performance caused by the body and chassis vibrations. These vibrations have a negative impact on the strength of a vehicle and its assemblies. To prevent undesired consequences of the vibration on the vehicle’s body and chassis components, the vehicle suspension system is provided with a shock absorption system. The use of optimal physical and geometrical parameters for the suspension system is crucial for reducing body vibration [1]. The promotion of energy-efficient solutions for unmanned vehicles in industry has resulted in their miniaturization and has simultaneously inspired works on new types of drivetrains to be carried out.

Examples of these are the omnidirectional wheels [2] of mobile robots, as shown in Figure 1, which offer improved maneuverability. An omnidirectional wheel is generally a wheel with rollers around the circumference whose roller axes are tangent to the wheel’s circumference [3]. In a complicated working environment, it is essential for mobile robots to be flexible, adaptive, and safe [4,5,6]. Etisoft Smart Solutions Sp. z o.o. designs and manufactures highly specialized vehicles that are customized to suit a client’s needs, including AGV and AMR industrial vehicles (Figure 1). These vehicles are often fitted with unconventional, unique, and innovative solutions that are specific to the drivetrain. For instance, the AGV tractor unit shown in Figure 2 has omnidirectional wheels and is made for carrying baskets loaded with goods. These wheels provide support at the vehicle’s end axes, while the middle axis has directional wheels that are driven by a dedicated engine on each side. In this configuration, a differential drive is provided, and the rotation of the wheels without movement is possible. Casters were initially used as support wheels but were replaced with traditional omnidirectional wheels due to their large working area that has a nearly circular shape.

These wheels have a small working area and offer the possibility of driving in a transverse direction. The small working area saves space that would otherwise have been occupied by casters during maneuvers. However, additional cinematic excitation originating from the irregular envelope [7,8] makes this a disadvantage of the omnidirectional wheel. Due to the specific structure of the omnidirectional wheel, when the wheel is rotated we encounter successive rollers, which is a situation similar to driving over small obstacles. A description of the resulting problems can be found in the works [9,10].

The method employed in [11,12] for finding the parameters of an optimal structure involved conducting experimental model tests and numerical simulations. A numerical model enables an unlimited number of tests to be carried out for various initial conditions. The parameters of an applied model have a considerable impact on the reliability of the final solution. For tests on an absorption system that is installed in a vehicle’s suspension system and fitted with omnidirectional wheels (Figure 3), simulation precision is essential in order to capture the excitation originating from the irregular envelope of the omnidirectional circle [8]. Figure 3 shows the schematics of a vehicle’s half model, where the parameters Xo,1 and Xo,2 are the excitation parameters.

Properly tuned shock absorbers ensure that the performance of a suspension system is satisfactory by minimizing the forces generated by the wheel–surface contact [13]. A test stand was consequently designed to determine the cinematic excitation originating from omnidirectional wheels rolling on a surface, and tests on the suspension system of mobile robots were performed, particularly on the omnidirectional wheel systems.

Previous methods of testing the excitation generated by omnidirectional wheels that relate to vibration acceleration measurement for specified body components have been investigated [14,15]. In the literature, information on specialist stands designed for testing vehicle suspension systems with omnidirectional wheels is limited. The solutions specified in the literature are based on traditional wheels and the determination of resistance [16]. Earlier research on mobile robots that are fitted with omnidirectional wheels has covered topics such as the kinematic modelling of vehicle movement [17] and control methods [18,19]. The described results confirm that the use of omnidirectional wheels significantly increases the range of robot movements compared to the use of conventional wheels [20].

Other research papers have focused on the design of an optimal omnidirectional wheel, taking into account the strength or reduction in vibration generated by the wheel. Vertical vibration is affected by various factors, such as geometry errors, the gap and thickness of a flexible body, the angular velocity, the alignment angle, the load, and the elasticity of a flexible body [8]. To improve the strength and performance of an omnidirectional wheel, the authors of [21] proposed the use of an omnidirectional wheel of a differential type. The omnidirectional wheel was constructed optimally by minimizing the gap between the rollers to ensure a long contact with the ground’s surface and a possible reduction in vibration, whereas the authors of [15] empirically evaluated how a heavy-weighted mobile robot can take advantage of its passive suspension system in order to use non-optimal or suboptimal omnidirectional wheels with a non-optimized inner gap.

The performance of mobile vehicles fitted with omnidirectional wheels depends on the type of wheels used and their location. Omnidirectional wheels are widely employed in omnidirectional vehicles; however, their location is determined arbitrarily. The paper [22] suggests that the performance of an omnidirectional vehicle is impacted on by the location of Mecanum wheels. The results obtained focus on two commonly used types—namely, Type-X and Type-O.

The dynamic modelling of omnidirectional wheels is based on circular models related to a multipart system. For example, the authors of [23] presented contact tracking algorithms for omnidirectional wheels rolling on a horizontal surface. The proposed approach was employed in previous research to determine the wheel–rail traction point [24,25]. The proposed algorithms demonstrate the reliability of the system and the authors demonstrate an impact-less method of transferring contact points from one roller to another during wheel rolling. The work presented in [26] analyzes the dynamics and vibration of an orientation motion platform that utilizes a sphere actuated by omnidirectional wheels. This work indicates that the geometry of omnidirectional wheels has a significant effect on system performance.

The aforementioned papers assumed the ideal geometry of interfacing elements [27]. The geometry of an omnidirectional wheel and an industrial floor on which the wheel is rolling show some deviations from this assumption.

This paper presents a test stand for determining the irregular envelope of the omnidirectional circle. These tests can be carried out for different configurations of a wheel in relation to the direction of travel as well as for different speeds of the wheel being rotated. The determined waveforms reflect the geometric profile of the contact points of the rolling omnidirectional wheel on the selected type of ground and constitute a model of kinematic excitation used in modeling the movement of a vehicle equipped with omnidirectional wheels.

## 2. Test Stand Description

### 2.1. Working Principle

The Department of Theoretical and Applied Mechanics of the Silesian University of Technology in Gliwice designed a test stand for the purpose of testing a vehicle suspension upright system, particularly for the omnidirectional wheels of mobile robots. A patent application named “Device for testing of vehicle suspension upright system, in particular related to omnidirectional wheels and methods for testing of vehicle suspension upright system” was submitted to UPRP in 2020 under No. P. 432948, with the test stand as the subject matter.

The test stand was designed to measure the vibration transferred onto the suspension upright system generated by wheel–surface contact depending on the wheel geometry as well as the condition of the surface on which the wheel is rolling. Figure 4 illustrates the block diagram of the described test stand.

The operating principle of the device is based on forcing the movement of a suspension system or a single wheel, denoted as (6) in Figure 4, which is connected to the vertical upright element denoted as (5). The vertical upright element is powered by a linear drive denoted as (2) to drive a truck, which is represented as (1). The movement of the wheel’s suspension system, denoted as (6) on the horizontal aluminum surface (8), has different uniformity. The linear drive (2) is driven by an electric engine (3) coupled with a control system to regulate the speed of the truck (1) at the given acceleration or with certain braking characteristics. The truck, denoted as (1), is connected to element (4) and the vertical upright system (5), which enables free movement in a vertical plane due to the force generated by ground irregularities (8) or an irregular wheel envelope (6). The measurement of the upright vertical movement (5) in relation to the guide (1) that is connected to the reference system is fitted with a position sensor (9). The tested component, which is the omnidirectional wheel (6), may be subjected to a greater load using additional weights (7).

### 2.2. Drive System

In this investigation, the drive system used is the standard solutions supplied by Item Company. The test stand depicted in Figure 5 and Figure 6 includes the following electro-mechanical components:PC with a drive control card and MATLAB software;Control cabinet provided with interface TTL/+24 V;Linear drive with a 1 kW servo motor installed at its end;Vertical upright with an installed omnidirectional wheel.

Figure 5 shows the connection of the particular components. For example, the red arrows demonstrate flow of energy/information from one medium to the other, whereas the blue arrows show the drive transmission of particular elements on the *x* and *y* axes. The horizontal displacement of the truck’s driving wheel is on the *x*-axis, while the direction of the upright free movement generated by the wheel–ground contact is on the *y* axis. The symbols φ*_z_* and φ*_y_* represent the circle rotation angle and the rotation angle of the wheel in relation to the vertical upright axis, respectively. The rotation angles are within the range of 0° to 90°.

The speed of the truck is controlled via an application in the MATLAB software. Control signals between the computer and field device are transmitted through a RT-DAC4/PCI real-time card connected to a signal conditioning interface. A control unit fitted with an engine servomechanism provides an additional element that is required for the proper operation of the device. As the drive system is based on speed control, the servomechanism provides the engine with a torque controller to ensure that the desired rotational speed is achieved [26]. The control unit receives transmitted control signals based on desired movement, after which the drive engine’s settings are programmed.

### 2.3. Measuring System

As shown in Figure 6, the following sensors were used by the measuring system during the testing to determine the excitation generated by the omnidirectional wheel when in motion:Two optical sensors, Philtec RC171 (A and B);One linear displacement sensor, LVDT Solartron Metrology (Figure 6C).

All instrumentation used for taking measurements utilized an analogue output signal within the range of 0–10 V and were connected by an RS 232 to ESAM Traveler 1 measuring device. All measurements were taken and recorded in real time by the PC computer.

The first optical sensor (Figure 6A) gives measurements of the upright vertical motion generated by the wheel–ground contact. Excitation parameters are determined based on the measurement results. The second optical sensor (Figure 6B) provides measurements of the wheel rotation angle. The parameters of the hardware devices used are presented in Table 1.

The construction of the tested wheel with a fixed rotation axis has made it impossible to use a traditional rotary encoder. Consequently, a unique solution for measuring the angle of rotation was proposed. The proposed solution required using a distance linear sensor. For this purpose, a special measuring plate whose facial surface feature had a 4 mm pitch was designed and manufactured with a 3D printer.

The CAD model of this measuring plate is depicted in Figure 7. By taking linear measurements, the following parameters can be determined: angle of rotation, gain of angle of rotation, angular speed, and precise positioning of particular rollers around the circumference of the omnidirectional wheel.

Figure 8 presents plots of the distance recorded by the measuring system per function of time at angles 0° and 60°, respectively. Further details are provided in Section 3. The visible imprecision (assumed linear performance) resulting from the method employed to manufacture the plate is due to the imprecision of 3D printing based on Fused Deposition Modeling (FDM). This imprecision mainly specifies the surface’s nature, which is made up of layers placed at a 0.15 mm resolution. The measured readings were observed to be identical for repeated tests performed in the same conditions, indicating the test’s repeatability and the absence of wheel slippage during movement at the same time. Undoubtedly, using a plate made from Computer Numerical Control (CNC) machined metal would ensure more precise measurements. However, for the purposes of this test, the precision of the measurements obtained was sufficient. From analyzing the readings, the wheel slippage during the wheel’s motion could be determined. The reliability of readings is essential, particularly because of the envelope irregularity of the omnidirectional wheel. Furthermore, it is possible to determine reference positions of particular rollers around the circumference of the omnidirectional wheel based on the obtained readings.

As it was impossible to ensure that the rolling of the omnidirectional wheel on the ground showed features with a perfect stiffness, the linear displacement sensor was used to compensate for the distortion of upright displacement measurements. Figure 9 shows a plot of the ground distortion along the measuring length where the omnidirectional wheel was rolled. The aluminum section and industrial floor were covered with PVC vinyl flooring.

To determine the irregularity of the omnidirectional wheel envelope, the vertical upright beam was set in motion with the linear drive so that the speed of the upright beam and omnidirectional wheel was sufficient for measurements to be taken. Measurements of the upright vertical displacements in relation to the truck moving horizontally were then taken. It is important to note that there was no vertical truck displacement. Upright vertical displacements resulted from the irregular envelope of the omnidirectional wheel and the irregularity of the surface that the wheel was rolled on. To eliminate any interference caused by surface irregularities, a smooth aluminum section supported on both ends by three conical adjustable supports was used. Measurements of deflection caused by wheel load were taken nonetheless due to the possible deformity of the section to compensate for the other measurements related to the wheel. The upright vertical displacement generated by the wheel rotation was measured with a displacement sensor coupled with a processing and recording system, and the data collected were archived. Controlling the speed settings of the linear drive allowed an adequate control of the wheel’s speed to be achieved, which had an impact on the performance of the forced movement resulting from the wheel–surface contact.

## 3. Performance of Tests

To determine the excitation generated by the omnidirectional wheel, a test was performed in quasi-static conditions at a very low rotational speed of the wheel. For the test, the traditional Rotacaster-branded 125 mm-diameter omnidirectional wheel was used, which included two rows of passive rollers. The technical specification of the Rotacaster omnidirectional wheel is given in Table 2. During testing, the wheel was subject to a load of 80 N, which corresponds to the static force that presses the wheel against the surface in AGV vehicles. Measurements of the irregularity of the omnidirectional wheel envelope were taken for seven variants at different wheel angular positions (φ*_y_* angle indicated in Figure 5) in relation to the movement direction at 0°, 15°, 30°, 45°, 60°, 75°, and 90°. When the vehicle makes a turn, a 90° angle relates to the rotation around its pivot. The vehicle turn is a result of the drive’s wheel differential speed. The upright element in all variants was driven at a constant speed *V_x_* = 6.72 mm/s (approx. 0.007 m/s).

Readings for the irregularities of the omnidirectional wheel envelope at variants I-VI were specified with the wheel’s circumference distance function for one total rotation of the wheel. As there was no rotation of the wheel at a 90° angle, where variant VII and the roller are rolling only, readings of the roller envelope irregularities were presented with the upright linear displacement function. In Table 3, the presented irregularity of the omnidirectional wheel envelope was determined by an initial reading deduction from measurements taken for the upright displacement. To compensate for the deflection caused by rolling the wheel, the value of the displacement caused by deflection that had been caused previously was deducted.

## 4. Discussion

The following observations and conclusions can be gained from analyzing the obtained results:Results regarding the irregularities of the omnidirectional wheel envelope are distinct for particular variants. In general, readings for the same variants were repeatable, as particular values of the envelope irregularity depended on changing the wheel’s geometry or stiffness and the variable geometry or deformability of particular rollers (Table 3, variant VII) resulting from the roller manufacture method, for instance. The final irregularity of an omnidirectional wheel also depends on the initial position of the roller in contact with the surface.The obtained wheel irregularity differs from the geometry profile, as it considers the deformability of rollers and their assembly.According to the analysis of the readings displayed in Table 3, for a wheel at an angle of 0°, particular rollers are in contact with the surface one after the other. For a wheel at an angle of 15–45°, two rollers are in contact with the surface simultaneously, and three rollers for a wheel is at an angle of 60–75°. In each case, there was no continual contact between the wheel and the surface.

The presented analysis of the readings suggests that the application of an optimal scenario based on the contact geometry is unreasonable for modeling the omnidirectional wheel–surface interaction. The variance in geometry parameters and manufacturing precision—i.e., geometry model/real object—often shows that the suitability of model-related considerations depends on the simulation precision of the wheel and surface interaction. It is, therefore, vital for the wheel to roll on an industrial floor instead of a perfectly smooth surface.

The results discussed in this paper have some limitations. The kinematic excitation we determined in the manuscript was related only to the irregular circumference of the circle and, therefore, it was necessary to take measurements at low velocities. Unfortunately, the vibration characteristics of a wheel moving at higher speeds are not directly proportional to motion at low speeds. This is due to many factors, most notably inertia. The design of the vehicle is also of great importance, as it influences the weight distribution and determines the suspension characteristics [28].

In addition, in vehicles the omnidirectional wheels are separated from the body by a damping system that also influences the behavior of the wheel. Therefore, it is very difficult to find a universal function that describes the override of the omnidirectional wheel. Thus, the characteristics presented in the paper are a certain approximation of the actual excitations, but they can be used in model tests—e.g., for the optimization of vehicle suspensions.

## 5. Conclusions

This paper has presented a measuring system developed for the determination of kinematic excitation originating from an irregular envelope of an omnidirectional wheel. A unique method was described for measuring the angle of rotation of the omnidirectional wheel based on a specially designed plate that was installed in the wheel and an optical sensor used for distance measurement. The readings indicated that it is unreasonable to apply virtual geometrical data (obtained in the design phase) when modeling. The application of optimal geometrical parameters for such systems while ignoring the inaccuracy of manufacturing particular elements, as well as the material used, typically results in false readings of the kinematic excitation originating from the rolling of the omnidirectional wheel. When considering the dynamics of a vehicle while making a turn, the model tests should take into account appropriate excitation parameters for such a case. As shown in Table 3, the excitation profile resulting from the irregular envelope of the omnidirectional wheel is closely related to the angle at which the wheel is situated in relation to the moving vehicle. This characteristic is different when the vehicle moves in a straight line (variant I, Table 2) and different when the vehicle turns (other cases).

The designed stand also allowed the verification of the suspension system’s performance. Modeling tests were carried out to determine the optimal position of the suspension system and obtain respective results from the optimal parameters of the suspension system. A prototype of the suspension system (Figure 10) was developed and the obtained solution was verified through numerical simulations based on actual readings.

## Figures and Tables

**Figure 1 sensors-21-06931-f001:**
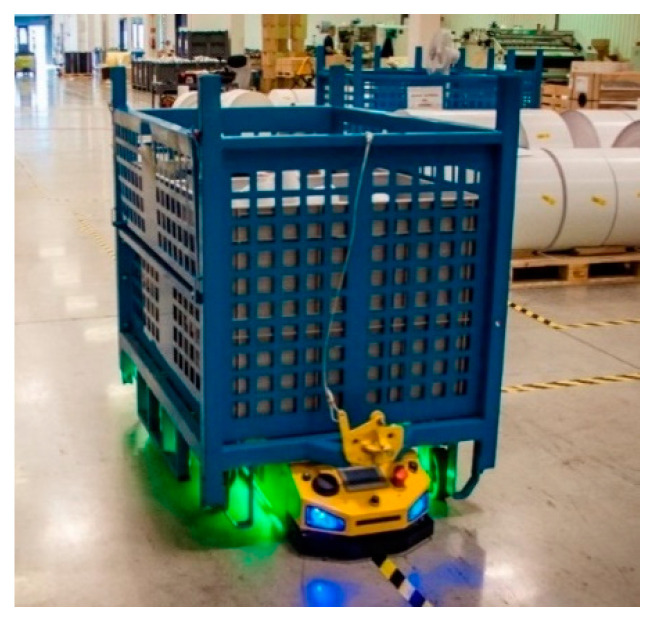
Tractor-type AGV Pull500 manufactured by Etisoft Smart Solutions Sp. z o.o.

**Figure 2 sensors-21-06931-f002:**
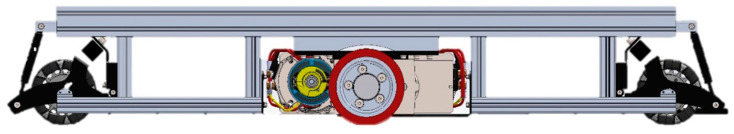
Lateral view of a vehicle frame with a drivetrain and drive system.

**Figure 3 sensors-21-06931-f003:**
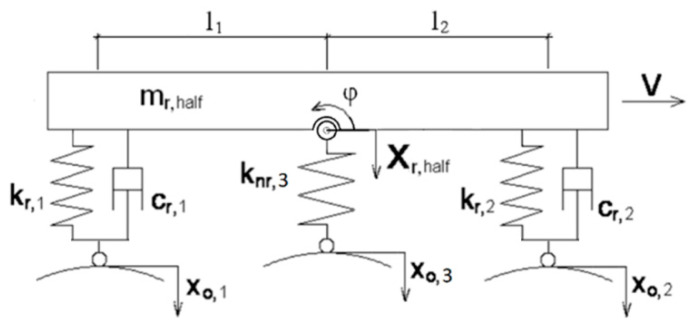
Half model of a vehicle.

**Figure 4 sensors-21-06931-f004:**
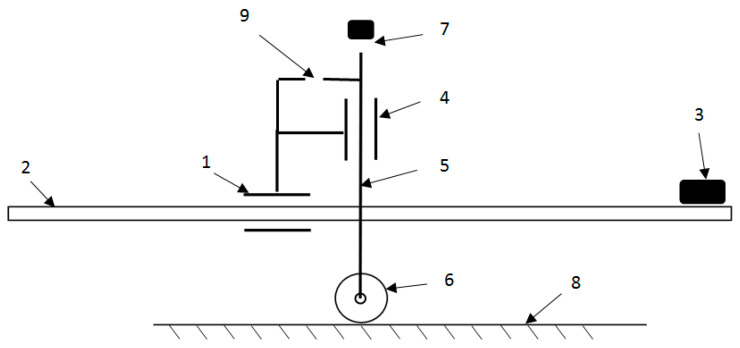
Designed test stand.

**Figure 5 sensors-21-06931-f005:**
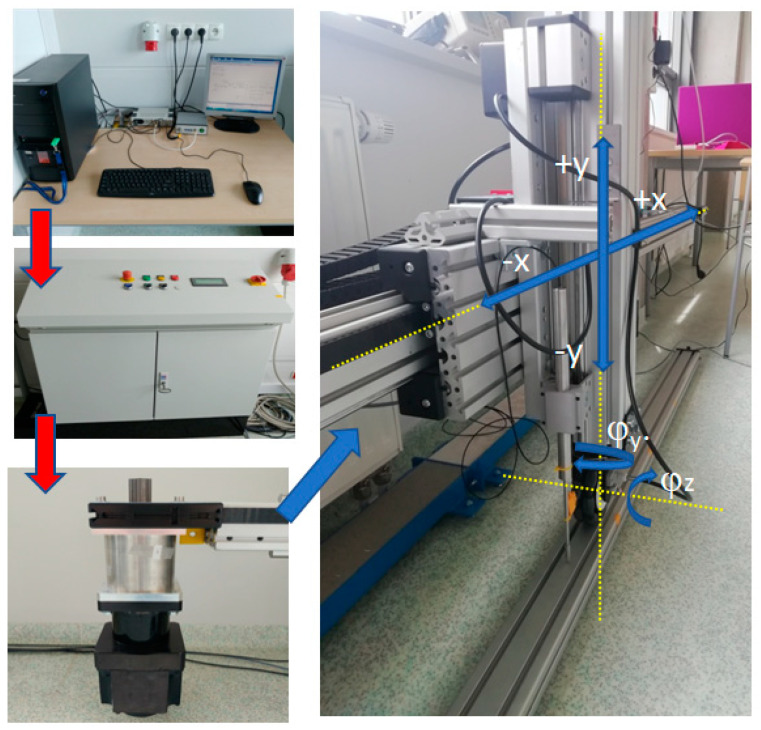
Block diagram showing the connection of the particular components.

**Figure 6 sensors-21-06931-f006:**
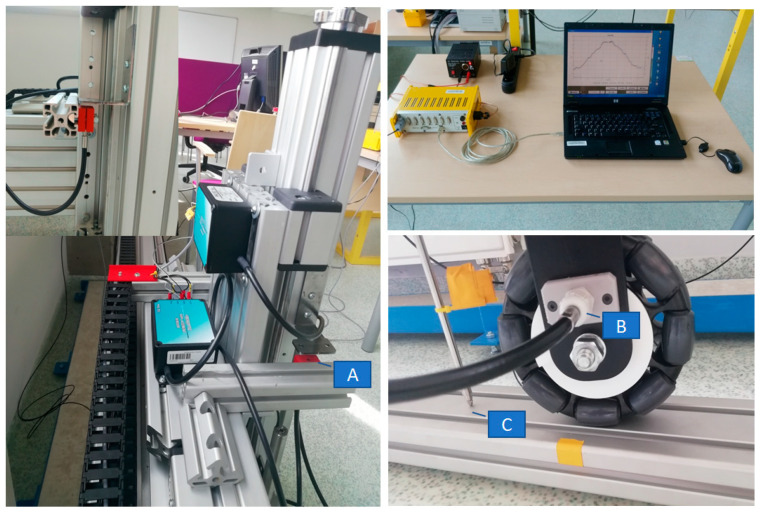
Position of the measuring instrumentation. (**A**) optical sensors, (**B**) optical sensors, (**C**) LVDT displacement sensor.

**Figure 7 sensors-21-06931-f007:**
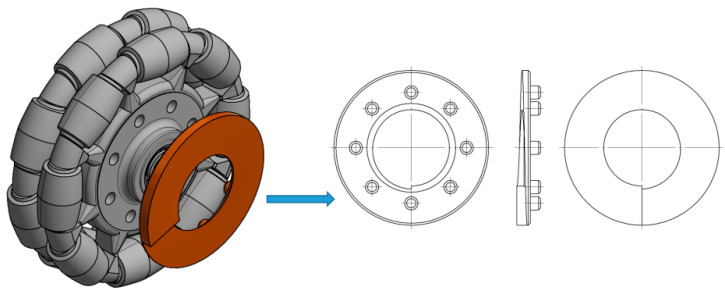
CAD model of omnidirectional wheel with measuring plate.

**Figure 8 sensors-21-06931-f008:**
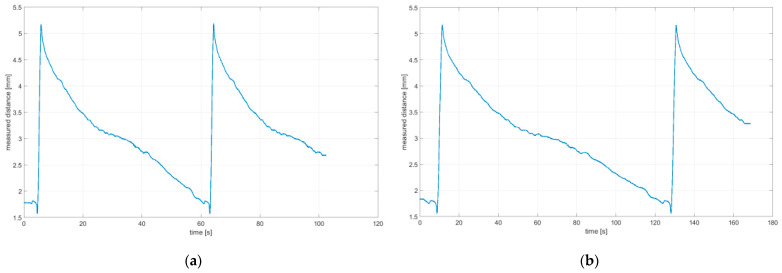
Distance measurement per time function: (**a**) 0°; (**b**) 60°.

**Figure 9 sensors-21-06931-f009:**
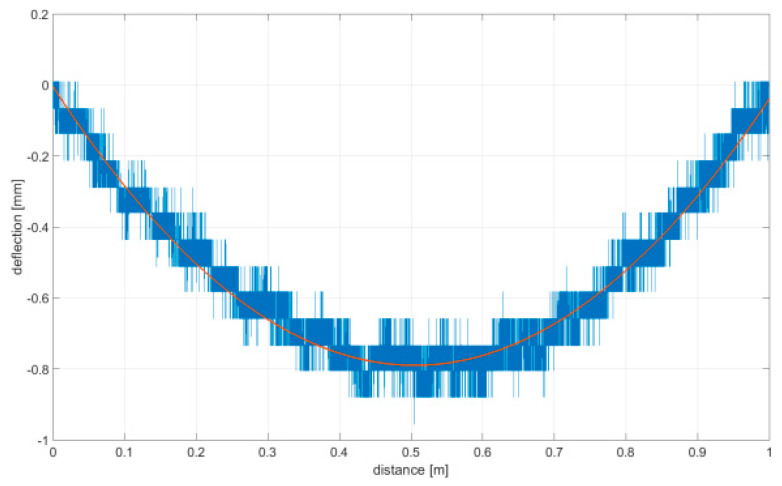
Deflection measurements (aluminum beam) are given in blue color; measured path interpolated with a polynomial function of 2 degrees is shown in red color.

**Figure 10 sensors-21-06931-f010:**
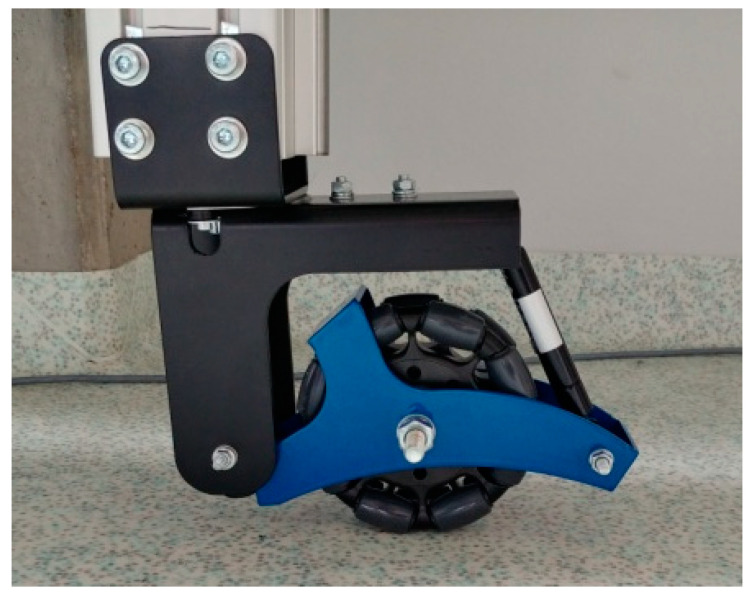
View of the vertical upright with the installed suspension system.

**Table 1 sensors-21-06931-t001:** Technical specifications of the Rotacaster 125 mm omnidirectional wheel.

Hardware Device	Parameters
optical sensors Philtec RC171	Light Source: LED, 850 nm; Input Voltage: +12 VDC; Analog Output: 0–5 V; Operating Range: 12.7 mm; Max. resolution: 1.3 µm.
S-Series displacement sensor LVDT Solartron Metrology	Measuring range: 150 mm; Analog Output: 0–10 V; Linearity FSO < 0.2%.
ESAM Traveler 1	16 bits A/D-converter; Sampling rates: 100 kHz; Signal conditioner/amplifier plug-in modules for: strain gauges, potentiometric sensors, piezo-resistive sensors, thermocouples, piezo-electric sensors according to ICP-standard, digital sensors, high-level voltage signals up to ±40 V, and other sensors.
PC with real time boards RT-DAC4/PCI	Analog inputs: 32 channels ± 10 V, resolution 12 bit, conversion time 1.6 µs; Analog outputs: 8 channels ± 10 V, resolution 12 bit, settling time 6 µs.

**Table 2 sensors-21-06931-t002:** Technical specification of the Rotacaster 125 mm omnidirectional wheel.

Type	125 mm Rotacaster
Catalogue number	R2-1258-95/S10
Passive rollers quantity	8
Wheel outer diameter	125 mm
Max roller diameter	20 mm
Roller radius of curvature	62.5 mm
Wheel static/dynamic load capacity	68/125 kg
Roller hardness	95 Shorea
Rollers	Polyurethane
Main bearings	2 × ball bearing
Roller bearings	2 × Nylon sleeves per roller
Body	Plastic
Distance between roller rows	19 mm
Distance between body and roller contact with surface	Nominal 2.5 mm, min. 0.5 mm
Wheel width	43 mm
Assembly hole diameter	10 mm
Weight	0.315 kg

**Table 3 sensors-21-06931-t003:** Irregularity of the omnidirectional wheel envelope. *w_z_*, wheel angular speed (rad/s).

Variant	Irregularity Graph of Omnidirectional Wheel Envelope
I-0°, *w_z_* = 0.1075 rad/s 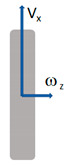	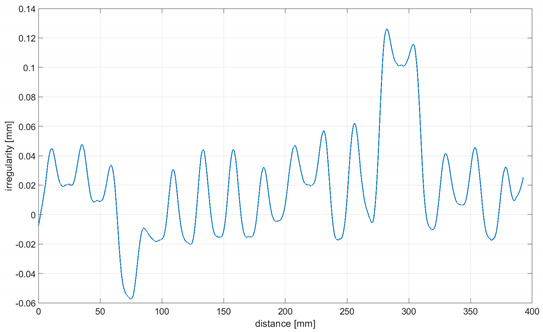
II-15°, *w_z_* = 0.1057 rad/s 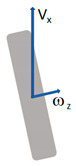	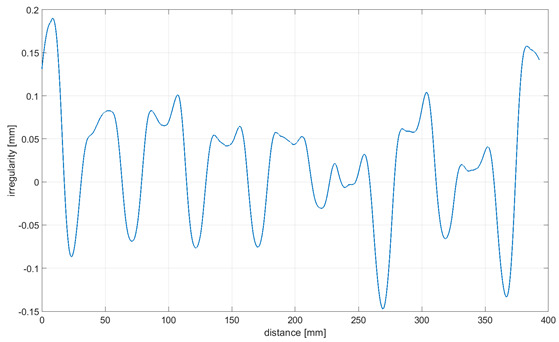
III-30°, *w_z_* = 0.0943 rad/s 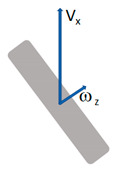	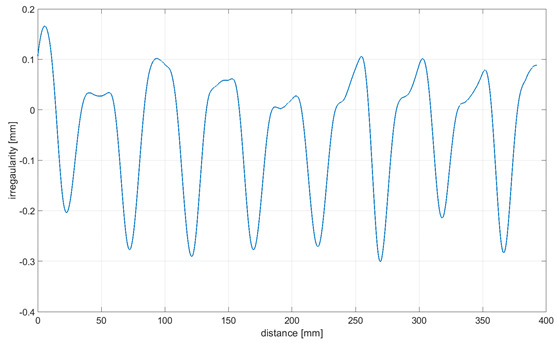
IV-45°, *w_z_* = 0.0757 rad/s 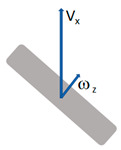	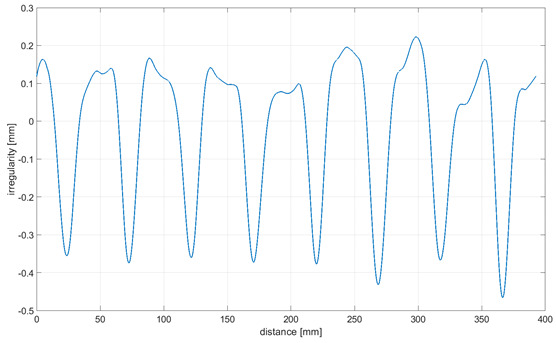
V-60°, *w_z_* = 0.0527 rad/s 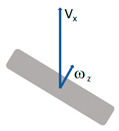	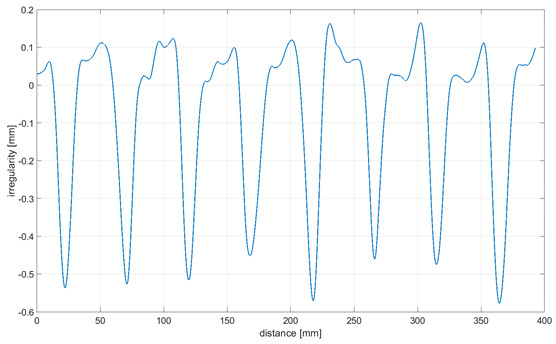
VI-75° *w_z_* = 0.019 rad/s 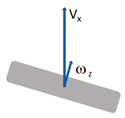	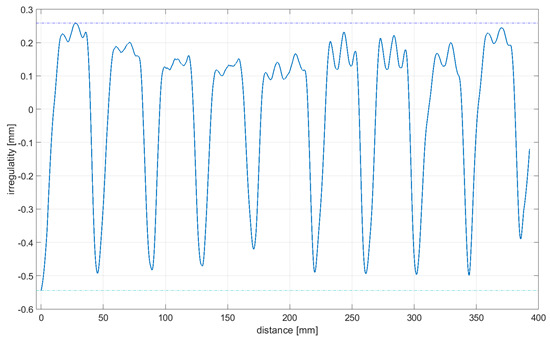
VII-90° *w_z_* = 0 rad/s 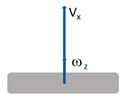	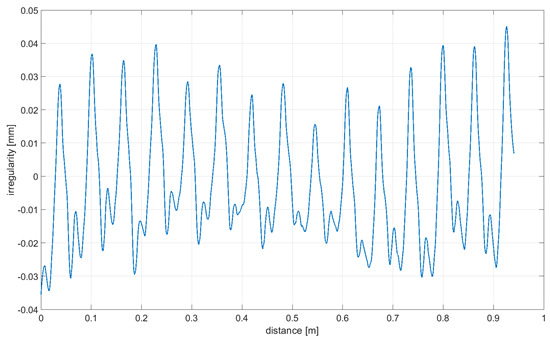

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
