# Peer review of "Determination of the Kinematic Excitation Originating from the Irregular Envelope of an Omnidirectional Wheel"

_sensors, 2021, doi:10.3390/s21206931_

Round 1
Reviewer 1 Report
This paper presents a test stand for determining the kinematic excitation originating from contact between a vehicle’s wheel and the ground. This excitation is unique to the movement of the omnidirectional wheel and originates from the irregular envelope of the wheel. The authors are suggested to address the following issues and improve the quality of the manuscript:
- What are the main contributions of this study?
- The authors need to introduce the parameters of the hardware devices (for example sensors, PC, actuator…etc.) and put them into a table so readers can easily follow, in this study.
- Did the authors evaluate this excitation when the vehicle was moving at different speeds? What is the maximum speed? in my opinion, the velocity of the vehicle also affects the excitation.
- In table 2, what does “Przebieg nieregularnoĹ›ci” mean?
- In the conclusions, the authors have written as “When considering the dynamics of a vehicle while making a turn, the model tests should take into account excitation parameters appropriate for such case (see Table 2)”, Can the authors explain more about this content?
Author Response
The authors would like to thank the Reviewer for appending the reviewers’ comments on our work that helped to improve the quality of the paper. According to the reviewer comments, the manuscript has been revised to incorporate those suggestions and a point-by-point reply to comments is given below. The changes ware marked in manuscript with yellow highlights.
Point 1:
“What are the main contributions of this study?”
Response 1:
In response to this important question regarding the contribution to research on model studies of the dynamics of the movement of vehicles based on omnidirectional wheels, the authors decided to redraft the final part of the introduction.
Point 2:
“The authors need to introduce the parameters of the hardware devices (for example sensors, PC, actuator…etc.) and put them into a table so readers can easily follow, in this study.”
Response 2:
As suggested, the list of devices used and the description of their parameters are given in a new table 1.
Point 3:
“Did the authors evaluate this excitation when the vehicle was moving at different speeds? What is the maximum speed? in my opinion, the velocity of the vehicle also affects the excitation.”
Response 3:
Omni-directional wheels are most often used in industrial vehicles that travel at low speeds. The research we carried out was part of the development work on the AGV mobile platform, which can move at a maximum speed of 5 km/h. Of course, we performed tests for different speeds. Indeed, as the reviewer assumed, the vibration characteristics of the wheel at higher speeds are not directly proportional to motion at low speeds. It is influenced by many factors, especially inertia, and this is the inertia of the entire vehicle. In vehicles, the omnidirectional wheels are separated from the body by a shock absorption system, which also affects the wheel behavior, so finding a universal function describing the excitation of the omnidirectional wheel is very difficult. The kinematic excitation we determined in the manuscript was related only to the irregular circumference of the circle, and therefore it was necessary to measure at low velocities. The characteristics presented in the paper are of course some approximation of the actual excitations, but they can be used in model tests to optimize vehicle suspensions. Importantly, the characteristics presented in the paper are the first published attempt to investigate such extortions and we hope that they will influence the further development of this field. A description of these restrictions is included in the section with the discussion of the results.
So thank you for this comment, as it shows the further direction of work on this issue.
Point 4:
“In table 2, what does “Przebieg nieregularnoĹ›ci” mean?”
Response 4:
We thank the reviewer for reading the text carefully. This mistake in translation has been corrected.
Point 5:
“In the conclusions, the authors have written as “When considering the dynamics of a vehicle while making a turn, the model tests should take into account excitation parameters appropriate for such case (see Table 2)”, Can the authors explain more about this content?”
Response 5:
In order to present the discussed conclusions more precisely, it was decided to add an additional description in the last chapter of the manuscript.
The authors would like to thank the Reviewer once again for valuable comments. We hope that the introduced changes have increased the quality of the revised paper.
Reviewer 2 Report
The presented issue is up-to-date and allows for inference about the behavior of the wheel and suspension on the basis of laboratory tests. The added value is the designed test stand allowing the testing of different omnidirectional wheels. The introduction lacks a slightly broader view of the kinematic excitations resulting from the tire-surface cooperation, and additional excitations related to obstacles appearing on the road, e.g. railway crossings, ruts, humps or random obstacles. / DOI 10.20858/tp.2016.11.1.11; DOI 10.1051/matecconf/201925404002 / Some obstacles can also be found in assembly / industrial halls, which is related to the possibility of moving cranes, with assembly channels or other installations. The next development stage should be the recognition of the range of noise emitted to the environment related to the cooperation of the omnidirectional wheels with the road surface.
In the work, it would be worth noting the impact of design changes and modernization of vehicles, with particular emphasis on tires and suspension. This issue is particularly important when designing new vehicles, developing a methodology for road tests of existing cars, developing criteria for the required scope of safety, and for assessing and assessing safety. The research methodology of vehicle motion parameters, depending on the selected trajectory, is described in standards. Regardless of road tests, laboratory and numerical tests are carried out.
1. Jun Zhu J., Khajepour A., Spike J., Chen S.-K., Moshchuk N., An integrated vehicle velocity and tire – road friction estimation based on a half-car model, International Journal of Vehicle Autonomous Systems 13 (2): 114-139, August 2016, DOI: 10.1504 / IJVAS.2016.078763
2. Klockiewicz Z., Ślaski G., Spadło M., Simulation Study of the Method of Random Kinematic Road Excitation's Reconstruction Based on Suspension Dynamic Responses with Signal Disruptions, Vibrations in Physical Systems - 2019, vol. 30, no. 2, pp. 2019 208-1-2019 208-8
3. Stanislav Evtukov, Egor Golov, Adhesion of car tires to the road surface during reconstruction of road accidents, E3S Web Conf. 164 03022 (2020), DOI: 10.1051 / e3sconf / 202016403022
4. Xiuyu Liu, Qingqing Cao, HaoWang, Jiaying Chen, and Xiaoming Huang, Evaluation of Vehicle Braking Performance on Wet Pavement Surface using an Integrated Tire-Vehicle Modeling Approach, Journal of Transportation Research Record, Transportation Research Board 2019, p. 1-13 , DOI: 10.1177 / 0361198119832886
5. DiĹľo, J .; Blatnický, M .; Sága, M .; Harušinec, J .; Gerlici, J .; Legutko, S. Development of a New System for Attaching the Wheels of the Front Axle in the Cross-Country Vehicle. Symmetry 2020, 12, 1156
6. Baumann R., Measuring Vehicle Dynamice with a Gyro Based System, Vehicle Dynamics & Simulation, Society of Automotive Engineers, Inc. 2003, p. 9-15
7. Sharifzadeh M., Timpone F., Farnam A., Senatore A., Akbari A., Tire-Road Adherence Conditions Estimation for Intelligent Vehicle Safety Applications, In book: Advances in Italian Mechanism Science Publisher: Springer International Publishing, November 2017, DOI: 10.1007 / 978-3-319-48375-7_42
8. Kulikowski K., Szpica D., Determination of directional stiffnesses of vehicles ’tires under a static load operation, Maintenance and Reliability 16 (1): 66-72, 2014
9. Dabrowski K., Ĺšlaski G., Method and algorithm of automatic estimation of road surface type for variable damping control, IOP Conference Series-Materials Science and Engineering, Vol. 148, 2016, DOI: 10.1088 / 1757-899X / 148 / 1/012028
Literature
Self-citations 4/26
11/26 from before 2016.
5/26 from before 2011.
2/26 from before 2000.
Authors should try to ensure that the self-citations do not exceed 10%.
Authors should try to ensure that the literature older than 10 years does not exceed 25%.
Author Response
The authors would like to thank the Reviewer for appending the reviewers’ comments on our work that helped to improve the quality of the paper. According to the reviewer comments, the manuscript has been revised to incorporate those suggestions and a point-by-point discussion/reply to comments is given below. The changes ware marked in manuscript with yellow highlights.
Point 1:
“The presented issue is up-to-date and allows for inference about the behavior of the wheel and suspension on the basis of laboratory tests. The added value is the designed test stand allowing the testing of different omnidirectional wheels. The introduction lacks a slightly broader view of the kinematic excitations resulting from the tire-surface cooperation, and additional excitations related to obstacles appearing on the road, e.g. railway crossings, ruts, humps or random obstacles. / DOI 10.20858/tp.2016.11.1.11; DOI 10.1051/matecconf/201925404002 / Some obstacles can also be found in assembly / industrial halls, which is related to the possibility of moving cranes, with assembly channels or other installations. The next development stage should be the recognition of the range of noise emitted to the environment related to the cooperation of the omnidirectional wheels with the road surface.”
Response 1:
Thank you very much for this remark that gave us a lot to think about. Of course, the issues raised in this remark are extremely important in the context of improving traction parameters. However, we believe that a detailed description of issues related to, inter alia, with tire-surface contact would introduce some inaccuracies as the omnidirectional wheels are tireless wheels. Therefore, we have not extended the introduction to this issue. However, we added an annotation that the phenomena occurring in the discussed issue correlate with the problem of crossing obstacles. We wanted to mention that our stand allows for the study of the impacts while driving through the dilatation, but in the submitted manuscript we tried to eliminate the influence of external factors and measure only the excitation from the idle wheel itself.
Point 2:
“In the work, it would be worth noting the impact of design changes and modernization of vehicles, with particular emphasis on tires and suspension. This issue is particularly important when designing new vehicles, developing a methodology for road tests of existing cars, developing criteria for the required scope of safety, and for assessing and assessing safety. The research methodology of vehicle motion parameters, depending on the selected trajectory, is described in standards. Regardless of road tests, laboratory and numerical tests are carried out(...)”
Response 2:
As the reviewer rightly noted, design changes affect the behavior of the suspension system. This remark is similar to that of another reviewer about the effect of speed on input behavior. In line with these remarks, the chapter with the discussion of the results has been edited. However, we would like to mention that, as in the previous note, we did not cover the issues related to the tires.
Of course, we are aware of the limitations of the results presented. The main goal of our research, however, was to examine only the kinematic excitations from the omnidirectional wheel, and the recorded signals will be used in further development work to optimize the shape of the AGV platform.
Point 3:
“Authors should try to ensure that the self-citations do not exceed 10%.
Authors should try to ensure that the literature older than 10 years does not exceed 25%.”
Response 3:
After revision of the manuscript and adding several important publications, the citation statistics complies with the recommendations..
The authors would like to thank the Reviewer once again for valuable comments which are a sure indication for further development work. We hope that the introduced changes are sufficient and increased the quality of the revised paper.